# Relationships Among Growth, Carcass Characteristics, and *Myf5 Myf6*, *MyoD,* and *MyoG* Genes Expression Level in Saanen Male Kids with Varying Slaughter Weights

**DOI:** 10.3390/ani15010016

**Published:** 2024-12-25

**Authors:** Uğur Şen, Dilek Gökçek, Ömer Faruk Yılmaz, Hüseyin Mert Yüksel, Hasan Önder, Emre Şirin, Sibel Bozkurt, Sezen Ocak Yetişgin, Ceyhun Yücel, Karlygash Omarova, Thobela Louis Tyasi

**Affiliations:** 1Department of Agricultural Biotechnology, Faculty of Agriculture, Ondokuz Mayis University, 55139 Samsun, Türkiye; dilekniyan27@gmail.com; 2Department of Animal Science, Faculty of Agriculture, Ondokuz Mayis University, 55139 Samsun, Türkiye; omer.yilmaz@omu.edu.tr (Ö.F.Y.); honder@omu.edu.tr (H.Ö.); sezen.ocak@omu.edu.tr (S.O.Y.); 3Department of Animal Science, Faculty of Agriculture, Erciyes University, 38280 Kayseri, Türkiye; huseyinmertyuksel@erciyes.edu.tr; 4Department of Agricultural Biotechnology, Faculty of Agriculture, Kırşehir Ahi Evran University, 40100 Kırşehir, Türkiye; emre.sirin@ahievran.edu.tr; 5Department of Animal Science, Faculty of Agriculture, Dicle University, 66200 Diyarbakır, Türkiye; sibel.b@dicle.edu.tr; 6Department of Animal Science, Faculty of Agriculture, Yozgat Bozok University, 21280 Yozgat, Türkiye; ceyhun.yucel@yobu.edu.tr; 7Department of Technology and Processing of Livestock Production, Faculty of Veterinary and Animal Husbandry Technology, S. Seifullin Seifullin Kazakh Agrotechnical University, Astana 010000, Kazakhstan; karlygash.omarova@mail.ru; 8Department of Agricultural Economics and Animal Production, School of Agricultural and Environmental Sciences, University of Limpopo, Private Bag X1106, Sovenga 0727, Limpopo, South Africa; louis.tyasi@ul.ac.za

**Keywords:** Turkish Saanen, MRF genes family, skeletal muscle, meat yield, kid

## Abstract

Practical and ethical problems arise in farms specializing in female-derived products (milk, eggs), as the evaluation of surplus male offspring is limited. Defining the carcass quality of male kids born from dairy goats, especially the Saanen breed, and knowing the factors affecting carcass yield is essential in improving carcass quality and increasing productivity. Therefore, it is necessary to understand the molecular basis of phenotypic diversity observed in terms of growth, carcass yield, and carcass characteristics in Saanen kids. This study aimed to determine the relationship between the expression levels of myogenic regulatory factors (MRFs) gene family members (*Myogenic Factor 5*, *Myogenic Factor 6*, *Myoblast Determination Factor,* and *Myogenin*) and growth and carcass characteristics in Saanen male kids with varying slaughter weights. The current results showed that changes in the expression levels of MRF gene family members may have significant and mixed effects on body weight and carcass characteristics of Saanen male kids raised and fattened under the same conditions.

## 1. Introduction

In goat milk production farms, female kids are mainly used as replacements, replacing older and poorer-performing goats to maintain the herd size and increase milk production capacity [1]. However, fewer male kids are raised to use in mating, resulting in a surplus of male kids. Most surplus male kids are either sold for meat production with low economic gain or killed. Therefore, the low financial value of surplus male kids also increases potential welfare concerns in the dairy goat industry [1]. Practical and ethical problems arise in farms specializing in female-derived products (milk, eggs), as the evaluation of surplus male offspring is limited [1]. Housing, health care, and slaughter cost significant money, and the income of surplus male offspring may not cover the costs. An alternative solution to the severe problem of surplus male kids is needed in dairy goat farms. Surplus male kids can be preferred to obtain meat in specific periods by applying various practices without compromising the existence of replacement male kids. Although the meat production potential of kids born from dairy goats is not as much as that of meat goat breeds, using surplus male kids in meat production can prevent waste of animal resources or ethical concerns. However, defining the carcass quality of male kids born from dairy goats and knowing the factors affecting carcass yield is essential in improving carcass quality and increasing productivity. Although studies on the growth, meat yield, and fattening performance of Saanen male kids have shown different phenotypic values [2,3,4,5] in Türkiye, the primary mechanism of phenotypic variation has not been revealed, and differences based on age, slaughter weight, breed characteristics, or environmental factors. Therefore, it is necessary to understand the molecular basis of this phenotypic diversity observed in terms of growth, carcass yield, and carcass characteristics in Saanen kids.

Myogenic regulatory factors (MRFs) serve as the primary regulators of skeletal muscle myogenesis and regulate the formation, differentiation, development, and proliferation of skeletal muscle precursor cells in the fetal period by expressing their encoded genes [6]. Additionally, MRFs regulate the development and functions of muscle fibers in the postnatal period [6,7,8]. The MRF gene family consists of four conserved basic helix–loop–helix (bHLH) transcription factors, *Myogenic Factor 5* (*Myf5*), *Myogenic Factor 6* (*Myf6*), *Myoblast Determination Factor* (*MyoD*), and *Myogenin* (*MyoG*) [6]. MRFs play an essential role in determining or regulating the expression of many genes [growth hormone and its receptor, *IGFs*, myostatin (*MSTN)*, myosin heavy chain isoforms] associated with economic characteristics such as postnatal growth, development, and meat yield [9]. Braun and Arnold [10] reported that *Myf5* and *MyoD*, from the MRFs family, are responsible for the development and proliferation of muscle fibers in the embryonic and fetal periods, and *Myf6* and *MyoG* are responsible for postnatal muscle maturation and development. However, studies have shown that *MyoD* may also be responsible for postnatal muscle fiber maturation by performing additional functions and that *Myf6* may be effective in the development and proliferation phase of the early formation process of muscle fibers [11]. Gene expression trends of MRF gene family members during the development and growth process have been determined in cattle [12], sheep [13], goats [9], and pigs [14], and muscle-specific expression levels have been defined and evaluated.

As a result, studies have reported that the MRF gene family significantly affects skeletal muscle development during the embryonic, fetal, and postnatal periods, resulting in alteration in meat production. Determination of the relationship between the expression levels of the MRF gene family and carcass characteristics in the Saanen breed, known as the dairy breed, can provide crucial information on the meat production potential and increasing meat yield of this breed. The results of this study may provide insight into the molecular mechanisms driving muscle development by highlighting the essential role of myogenic regulatory factors in the muscle development of Saanen kids with varying slaughter weights despite being reared under the same conditions. Therefore, this study aimed to determine the relationship between the expression levels of *Myf5*, *Myf6*, *MyoD*, and *MyoG* genes and growth and carcass characteristics in Saanen male kids with varying slaughter weights raised under the same conditions.

## 2. Materials and Methods

### 2.1. Animals

Twenty-eight singleton-bearing newborn male kids, born to Turkish Saanen goats in at least the second parturition and ranging from 2 to 3 years of age, were dried and weighed before suckling.

### 2.2. Growth and Fattening of Kids

All the kids were kept in the pen with their dams for two weeks and allowed to suckle their dams freely to ensure they received sufficient colostrum and milk. The goats were fed an average of 250 g concentrate (approximately 90% dry matter, 15% crude protein, and 2800 kcal/kg dry matter metabolizable energy) and alfalfa of 1 kg per day for two weeks. Two weeks after birth, goats were allowed to graze on pasture during the day and care for their kids overnight in the pen until weaned. In addition to pasture grazing, the goats were fed an average of 150 g of concentrate per day during lactation. When all kids were one month old, they were treated against internal and external parasites. Four kids died before one month of age due to various reasons. Traumatic lesions and infections were the primary cause of neonatal kid mortality (14.3%). To ensure adequate rumen development, the kids were also introduced to ad libitum concentrate diets for creep feeding and dry alfalfa in addition to their dam’s milk until weaning in the pen. At 120 days, all kids were weaned, and feed and water were withdrawn overnight to determine fasting body weight the following day at weaning. The traditional fattening practice was applied. After weaning, all kids were randomly divided into two groups of 12 animals and intensively fed ad libitum for 60 days in 20 square meter pens. Kids were fed ad libitum with double-sided galvanized feeders with adjustable shutters to regulate the flow of pelleted concentrate into the trough and an easy-open lid to allow easy filling during fattening. Also, alfalfa hay was introduced to the kids with a wooden hay feeder. Alfalfa hay and concentrates were offered to kids twice daily, in the morning (09:00) and evening (19:00). This feeding protocol aimed to ensure optimal nutrition and welfare throughout the fattening period. The nutrient contents of experimental feeds were provided by the manufacturer (Güven Yem A.Ş., Çorum, Türkiye) and shown in Table 1. Water and mineral stones were freely available during the fattening period. Four kids had to be removed from the fattening due to health problems. At the end of the fattening period, feed and water were withdrawn overnight to determine the fasting body weight of all kids the following day. The kids were divided into two groups: low slaughter weight (L; *n* = 11; ≤29 kg) and high slaughter weight (H; *n* = 13; >29) according to BW at the end of fattening.

### 2.3. Measurements and Muscle Sample Collection

All kids were transported to a local slaughterhouse, where they were stunned before slaughter. Standard commercial slaughter procedures were then carried out in accordance with ethical requirements. Immediately after slaughter, samples measuring 5 × 2 × 2 cm (approximately 50 g) were dissected from the mid-sections of Longissimus-dorsi (LD) and Semitendinosus (ST) muscles. Fat and connective tissues were trimmed from muscle samples, wrapped with sterilized aluminum foil, snap-frozen in liquid nitrogen, and stored at −80 °C until RNA isolation. The pelt, head, feet, internal organs (kidney, liver, spleen, heart, and lung), testis, empty reticulo-rumen, and empty intestine were weighed after slaughter. In addition, internal and kidney fats were isolated and weighed. The warm carcass was also weighed after removing all internal organs. The eviscerated carcasses were chilled for 24 h at 4 °C and reweighed to determine cold carcass weight.

### 2.4. Carcass Characteristics and Dissection of Carcass Parts

Before the chilled carcasses were cut into parts, the tail region was removed and then divided into right and left halves at the midline. Each side of the carcass was divided into six parts (shoulder, flank, hind leg, loin, rib, and neck) according to the methodology described by Colomer-Rocher et al. [17]. The percentages of carcass parts were calculated based on whole carcass weight. Both sides of the carcasses were used to estimate carcass composition. After carcass jointing, compositions of carcass parts were evaluated using the dissection method Fisher and De Boer [18] described. Each carcass part was dissected into muscle and bone tissues. In the dissection process, muscles with subcutaneous fat, intermuscular fat, and other tissues (connective, nerve, etc., tissues) were removed from bones. The muscle sample weight taken for RNA isolation from the LD and ST muscles was added to the final muscle weight of the loin and hind leg parts. Percentages of muscles (including fats and other tissue) and bone were calculated based on the weights of carcass joints before dissection.

### 2.5. Total RNA Isolation and Synthesis of cDNA

Total RNA of LD and ST muscle samples were isolated by a commercial RNA (PureLink ™, RNA Mini Kit, Invitrogen™, 12183018A, Waltham, MA, USA) purification kit using the TRIzol Reagent (Thermo Fisher Scientific, USA) as suggested by the manufacturer. Genomic DNA was eliminated by digestion with DNase I (Thermo Fisher Scientific Inc., Waltham, MA, USA). The purity and concentration of isolated RNA were evaluated by the A260/A280 ratio using a NanoDrop™ 2000/2000c spectrophotometer (Thermo Fisher Scientific Inc., Waltham, MA, USA), and all RNA samples showed A260/A280 values within the range of 2.01 to 2.08 and A260/A230 values above 2. The integrity of collected RNA was checked with 1% *w*/*v* agarose gel electrophoresis (Figure 1). The RNA samples were converted to cDNA using a commercial cDNA kit (BIORAD iScript cDNA, 1708890) following the manufacturer’s instructions in the Thermal Cycler (BIORAD, Hercules, CA, USA) device. The cDNA samples were further diluted to the same initial concentration (20 ng/μL) and stored at −20 °C until subsequent quantitative real-time PCR analysis.

### 2.6. qRT-PCR Analyses

Primers used for the amplification of genes were designed using online tools (https://www.ncbi.nlm.nih.gov/tools/primer-blast/) (accessed on 12 April 2022) based on the related gene sequences of caprine (Table 2). *GADPH* was selected as a housekeeping gene to normalize the expression of target genes. All primers were synthesized by Sentebiolab (Ankara, Türkiye). The specificity of each of the designed primers was checked via online Primer-BLAST (https://www.ncbi.nlm.nih.gov/tools/primer-blast/) and melt curve analysis was carried out during qRT-PCR. Relative quantification of all transcripts was performed by qRT-PCR using the CFX96 Touch real-time PCR Detection System (Bio-Rad Laboratories, Hercules, CA, USA). Real-time quantitative PCR was run with EvaGreen mastermix (5× HOT FIREPol EvaGreen qPCR Mix Plus, Solis BioDyne, Tartu, Estonya). The reaction mix was in a total volume of 10 μL comprising 5 μL of 5X HOT FIREPol mix, 0.5 μL of forward primer (10 μmol/L), 0.5 μL of reverse primer (10 μmol/L), Dye, 2 μL of DEPC treated water, and 2 μL of template cDNA. PCR amplification was carried out as follows: denaturation of 95 °C for 30 s, followed by 40 cycles of 95 °C for 5 s, and a specific annealing temperature of 60 °C for 30 s. The relative mRNA expression levels of the genes were calculated by the 2^−ΔΔCt^ method.

### 2.7. Statistical Analysis

The data on all the performance, slaughter, and carcass measurements were subjected to analysis of variance using the SPSS 17.0 (2008) package program (SPSS Inc., 2008, 17.0.1, Wacker Drive, Chicago, IL, USA). Significant differences between means were tested in analyses of variance with a completely randomized design. Shapiro–Wilk test results showed that all data were normally distributed (*p* > 0.05) and the Levene test results showed that the variances were equal (*p* > 0.05). Relationships between variable traits for data were determined with the Pearson correlation analysis. Results were computed as mean ± SE and statistical significance was determined at the level of *p* < 0.05.

## 3. Results

### 3.1. Body Weight Characteristics

Body weight characteristics of L and H kids at various ages are presented in Table 3. There were significant differences were observed between L and H kids concerning birth weight, weaning weight, and body weights recorded at 160, 170, and 180 days of age (*p* < 0.05). Also, there were no significant differences between L and H kids in average daily weight gain from birth to weaning, but H kids had higher average daily weight gain than L kids from weaning to 180 days of age (*p* < 0.05).

### 3.2. Carcass and Non-Carcass Component Characteristics

Carcass and non-carcass component characteristics are presented in Table 4. In this study, H kids had significantly heavier warm and cold carcasses compared to L kids (*p* < 0.05). However, no statistical difference was found between the kids in the H and L groups regarding warm and cold carcass yield. H kids had more head, foot, skin, kidney, liver, spleen, heart, and lung weight than L kids (*p* < 0.05). There was no statistically significant difference between kids in the H and L groups regarding kidney fat and internal fat weight. Likewise, although no statistically significant difference was observed between the slaughter weight groups concerning testicle and rumen weight, it was noted that H kids had significantly heavier small intestines compared to L kids (*p* < 0.05).

The characteristics of carcass parts of L and H kids are presented in Table 5. H kids had heavier shoulder, flank, hind leg, loin, rib, neck, and tail carcass parts than L kids (*p* < 0.05). Although carcass percentages of hind leg, rib, and tail parts in H kids were higher (*p* < 0.05), there were no significant differences between H and L kids regarding carcass percentages of shoulder, flank, loin, and neck parts.

Carcass dissection characteristics of L and H kids are presented in Table 6. The total weight of meat, fat, and connective tissue components of shoulder, flank, hind leg, loin, rib, neck, and tail carcass parts was higher in H kids than in L kids (*p* < 0.05). A similar trend was observed regarding bone weight in the same carcass parts (*p* < 0.05), except for the flank and loin parts. The percentage of muscle and bone in the shoulder, hind leg, and tail carcass sections exhibited a significant increase (*p* < 0.05) in H kids when compared to L kids. There were no significant differences in the percentage of bone in the flank between slaughter weight groups, but H kids had a higher (*p* < 0.05) muscle percentage in the flank compared to L kids. At the same time, there were no significant differences in terms of muscle and bone percentage of loin, rib, and neck carcass sections between slaughter weight groups. The total meat and bone weight of cold carcasses was significantly higher (*p* < 0.05) in H kids compared to L kids. There were no significant differences in bone percentage of cold carcass between slaughter weight groups, but H kids had higher (*p* < 0.05) muscle percentage in the flank compared to L kids.

### 3.3. Profiles of Myf5, Myf6, MyoD and MyoG Gene Expression

The study determined that the *Myf5*, *Myf6,* and *MyoD* genes expression level in the LD muscle of H kids was higher (*p* < 0.05) than that of L kids, but there were no significant differences in terms of the same genes in the ST muscle between slaughter weight groups (Figure 2). Additionally, *MyoG* gene expression levels of H kids were higher than those of L kids in both muscles (*p* < 0.05). As a result of quantitative real-time polymerase chain reaction (real-time qPCR), the expression levels of the *Myf5*, *Myf6*, *MyoD,* and *MyoG* genes of L and H kids were calculated as 3.46 ± 0.78 and 7.72 ± 0.70, 2.40 ± 0.49 and 6.84 ± 0.92, 2.26 ± 0.31 and 4.86 ± 0.37, 8.01 ± 0.85 and 12.75 ± 1.01 fold changes in the LD muscle and 3.16 ± 0.51 and 3.74 ± 0.39, 6.21 ± 0.85 and 5.28 ± 0.56, 4.48 ± 0.48 and 4.55 ± 0.47, and 3.47 ± 0.33 and 7.10 ± 0.50 fold changes in the ST muscle, respectively.

### 3.4. Correlation Coefficients Among the Slaughter Weight, Carcass Characteristics, and MRFs Gene Expression

The correlation coefficients among the slaughter weight, carcass characteristics, and MRF gene expression in the two major muscles are presented in Table 7. There were positive correlations (*p* < 0.05) among slaughter weight, cold carcass weight, the total edible meat content of carcasses, and MRF family member gene expression (except for *Myf6* and *MyoD* in ST muscle) in both muscles. In addition, positive correlations (*p* < 0.05) between loin weight and MRF family member gene expression in LD muscle were calculated. Moreover, positive correlations were found among hind leg weight and *Myf5* and *MyoG* gene expression in ST muscle (*p* < 0.05).

## 4. Discussion

Most phenotypes of economic importance are expected to be polygenic, with each gene making a small contribution to the phenotype. Identifying genes associated with phenotypes such as growth rate, fattening performance, and meat yield, unraveling their quantitative effects, and exploiting those with significant effects are essential to improve animal productivity. Growth rate is a polygenic trait of high economic importance, and in the present study, we focused on the relationship between growth and MRFs in Saanen male kids.

Studies investigating the relationships of specific candidate genes responsible for muscle development with phenotypic traits such as growth, meat yield, and carcass characteristics provide the new genetic evaluation of production traits for more reliable breeding practices in livestock. Growth rate and muscle development are considered essential traits in livestock, and muscle-specific transcription factors play a crucial role in meat production [19]. Therefore, the approach that the MRF gene family has a high potential for improving meat-related traits [20] may support the elucidation of the usability of male offspring in meat production in farms engaged in female-based output, where the use of excess male offspring is limited [1]. In the current study, carcass characteristics of similar-aged Saanen male kids with varying slaughter weights were compared, and the relationship between slaughter, carcass characteristics, and MRF gene family expression levels was examined. The present study demonstrates that MRF gene family expression levels in LD and ST muscles influence growth performance and carcass characteristics of Saanen male kids. Additionally, different slaughter weights of same-age Saanen kids raised under the same conditions may reflect the effects of MRF gene family expression levels. Te Pas [20] suggests that selection for increased growth rate is associated with increased MRF gene expression and regulation of satellite cell proliferation and differentiation, while selection for increased lean percentage is associated with increased muscle tissue maintenance.

There is a high correlation between growth rate and carcass yield. Moreover, the MRF gene family is highly associated with growth [9,20]. Therefore, in the present study, the effect of the expression patterns of the MRF family member genes on growth was evaluated. The results of this study shed light on the intricate differences between high- and low-weight kids in terms of various physiological and carcass characteristics. One of the striking findings in the study is that while there were significant differences in birth weight and weaning weight between H and L kids in the early stages of fattening (140 and 150 days of age), a possible normalization in growth trajectories is underlined and the difference disappears. Interestingly, in the middle and late stages of fattening (160, 170, and 180 days of age), the differences in body weight became more pronounced between H and L kids. This situation is further supported by the higher average daily weight gain in high-weight kids from weaning to 180 days, emphasizing the importance of post-weaning factors in influencing weight development. Our results indicate that MRF family member genes are associated with growth rates in Saanen kids due to H kids having higher expression levels of MRF family member genes.

In studies by Akdağ et al. [3] and Akbaş et al. [5] conducted on Saanen goats reared under similar conditions, the birth weights of Saanen kids were determined to be 4.04 kg and 3.22 kg, respectively. The birth weights determined for the Saanen kids in the present study were notably lower compared to those reported in previous studies [3,5], which documented different birth weights for similar genotypes. Conversely, the values obtained in this study were higher than those reported in previous studies [3,5] regarding the average body weight at various ages determined during the fattening period after weaning. Although H and L kids had similar daily weight gain from birth to weaning age, there was a significant difference between their growth rates after weaning until the end of fattening, mainly due to the live weight difference of 160, 170, and 180 days. Regrettably, the individual daily feed consumption of the kids could not be ascertained in the present study. However, the disparity in live weight observed during the final 30 days of the fattening period may potentially be linked to the diminished growth and feed efficiency of L kids. This phenomenon could be attributed to alterations in the expression of MRF genes, which play a pivotal role in postnatal development. Carcass composition analysis revealed significant differences between the slaughter weight groups in various organs and carcass sections. H kids displayed heavier warm and cold carcasses and weights of specific organs such as head, foot, skin, kidney, liver, spleen, heart, and lung, which were heavy. However, carcass yield was not affected by slaughter weight differences. The distribution of muscle, fat, connective tissue, and bone in different carcass parts further highlighted the complexity of weight-related variations. Supporting these results, previous studies [21,22,23,24] reported increased carcass weight with the increasing slaughter age or slaughter weight. Carcass yield did not change with the slaughter weight in the current study, as previously reported by Peña et al. [22], Panea et al. [25], and Yalcintan et al. [24]. Carcass yield is affected by the proportion of non-carcass components, specific organs, carcass fatness, and the content of the gastrointestinal tract [21,24]. The similarity in carcass yield in slaughter weight groups resulted from the parallel increase in carcass weights and the weight of non-carcass components and specific organs. In the current study, the carcass yield was calculated according to slaughter weight, thus preventing the carcass yield from being affected by the weights of specific organs and the gastrointestinal components. Similarly to the results of the current study, Teixeira et al. [26] reported that the carcass yield did not change between light and heavy kids, which had a carcass weight of 3.8–6 kg. Also, carcass yields of H and L kids were similar to previous studies’ reports for carcass yield in kids [22,23,25].

Transcriptional analysis can be an effective tool to characterize gene expression differences between muscle groups with different phenotypes [27]. Previous studies have reported consistent differences in the expression of genes by comparing muscle satellite cells in different muscles during the myogenic differentiation process; this suggests that differences in gene expression may play a role in directing muscle compartmentalization and development [27,28]. The expression of *Myf5*, *Myf6*, *MyoD*, and *MyoG* showed a significant difference in expression level in all investigated skeletal muscles. *Myf5* and *MyoG* gene expression in the LD muscle of H kids showed significantly higher levels in the ST muscle. Moreover, *MyoG* gene expression in the LD muscle of L kids exhibited substantially higher levels in the ST muscle. However, *Myf6* and *MyoD* gene expression in the LD muscle of L kids showed considerably lower levels in the ST muscle. The differences among *Myf5*, *Myf6*, *MyoD*, and *MyoG* gene expressions in LD and ST muscles may be related to the number of activated satellite cells in skeletal muscles [28,29].

Insufficient growth rates in kids slaughtered at low live weight cause poor carcass characteristics [30]. LD and ST muscles are considered predictors of the carcass meat characteristics and composition [31,32]. Hence, in our dataset, we evaluated the relationship among kids’ slaughter weight, carcass weights, and the expression patterns of MRF family member genes in LD and ST muscles. We focused on the results exhibiting higher correlation coefficient values for further evaluation. In LD muscles, the expression of the MRFs family member genes was positively correlated with slaughter weight, cold carcass weight, and carcasses’ total edible meat content. As expected, the expression of the MRFs family member genes in LD muscle positively correlated with the weight of the loin carcass join. Similarly, in the ST muscle, *Myf5* and *MyoG* gene expression were positively associated with slaughter weight, cold carcass weight, and carcasses’ total edible meat content. Moreover, in the ST muscle, significant positive correlations were observed among hind leg weight, *Myf5*, and *MyoG* gene expression. Although many reports have addressed how different exercises or stimuli affect MRFs and muscle fiber conversion, the specific molecular mechanisms governing muscle development remain unclear [8]. To our knowledge, this is the first study to report that despite being raised under similar conditions, alterations in carcass composition and body weights of Saanen male kids at the same age may be due to MRF gene expression patterns in different skeletal muscles. The positive correlations found among slaughter weight, cold carcass weight, total edible meat content, and muscle regulatory factor (MRF) gene expression in both LD and ST muscles further strengthen the link between weight and genetic factors. Additionally, specific correlations, such as those between loin weight and MRF family member gene expression in LD muscle, provide more nuanced insights into the relationship between carcass composition and gene expression.

It is widely accepted that muscle growth and development occur from both hyperplasia and hypertrophy. Feeding behavior [33,34], exercise intensity [35], and activity patterns [36] may influence the regulation of muscle development or meat quality. Some authors have previously reported that fiber type-specific differences are associated with MRFs. MRF gene expression patterns exhibit differences in different muscle masses due to differences in fiber types [37], nutrition [9,33,34], and exercise activity [38] in animals. The development of skeletal muscles is a highly coordinated process involving a highly complex signal regulatory network. Various molecules, including genes, miRNAs, and lncRNAs, have been found to participate in fetal and postnatal skeletal muscle growth and development through multiple signaling pathways and various ways. For example, the MRFs, paired box protein 3/7 (Pax3/7) [39], *MSTN*, and *myocyte enhancer factor 2* (*MEF2*) [40] family are key factors that regulate the development of muscles. Meanwhile, these factors can also be regulated by lncRNAs and some intracellular signaling proteins. MRFs control the differentiation of skeletal muscle cells during myogenesis, and the expression of MRFs can be significantly affected by nutrition [9,33,34] and exercise intensity [35]. In the current study, although all kids were raised under similar environmental conditions, significant differences were found in the expression of the MRFs gene family members *Myf5*, *Myf6*, *MyoG*, and *MyoD* were found between kids with different slaughter weights. This disparity may serve as an indicative measure of the growth potential exhibited by kids at the time of slaughter. These results may suggest that selection for increased growth rate is associated with increased MRF gene family members’ expression and regulation of satellite cell proliferation and differentiation [20]. The RNA concentration differences in some skeletal muscle may be due to alterations in the expression pattern of myogenic genes related to mutually interacting processes during muscle development [8,41]. The expression patterns of *Myf5*, *Myf6*, *MyoG*, and *MyoD* genes were more significant in especially the LD skeletal muscle of kids with high slaughter weight, suggesting that kids have more muscle mass and live weight by improving the development of muscle fibers. This situation was consistent with the fact that kids with higher slaughter weights have more outstanding total edible meat content and heavier carcass sections in their carcasses. This result also revealed the reason for the low carcass weight and total edible meat content in low-slaughter-weight kids, which was that the expression patterns of MRF genes in LD and ST skeletal muscles were low. qRT-PCR analysis of muscle tissue indicated higher expression levels of *Myf5*, *Myf6*, *MyoD*, and *MyoG* genes in high-weight kids’ LD muscle. This suggests a potential genetic basis for the observed weight differences, with specific genes associated with muscle development showing higher expression levels in the high-slaughter-weight group [8,9] and the positive correlation coefficients among the slaughter weight, carcass characteristics, and MRFs gene expression of this study also supported this association.

## 5. Conclusions

Our results suggest that changes in the expression levels of MRF gene family members may have significant and mixed effects on body weight and carcass characteristics of Saanen male kids raised and fattened under the same conditions. The study results showed that *Myf5*, *Myf6*, *MyoD*, and *MyoG* gene expression levels in the LD muscle, one of the major skeleton muscles, of Saanen kids with high slaughter weight were observed to be strikingly high. Moreover, the superiority in live weight and carcass traits of Saanen kids with high slaughter weight was attributed to this. This study contributes valuable information to understanding the complex interplay between weight, genetic factors, and carcass characteristics in growing animals. Further research in this area could provide insights into potential breeding strategies and management practices to optimize growth rate and meat quality in livestock. Hence, further studies are needed to determine the precise mechanisms involved in muscle development in animals raised under similar nutritional conditions.

## Figures and Tables

**Figure 1 animals-15-00016-f001:**
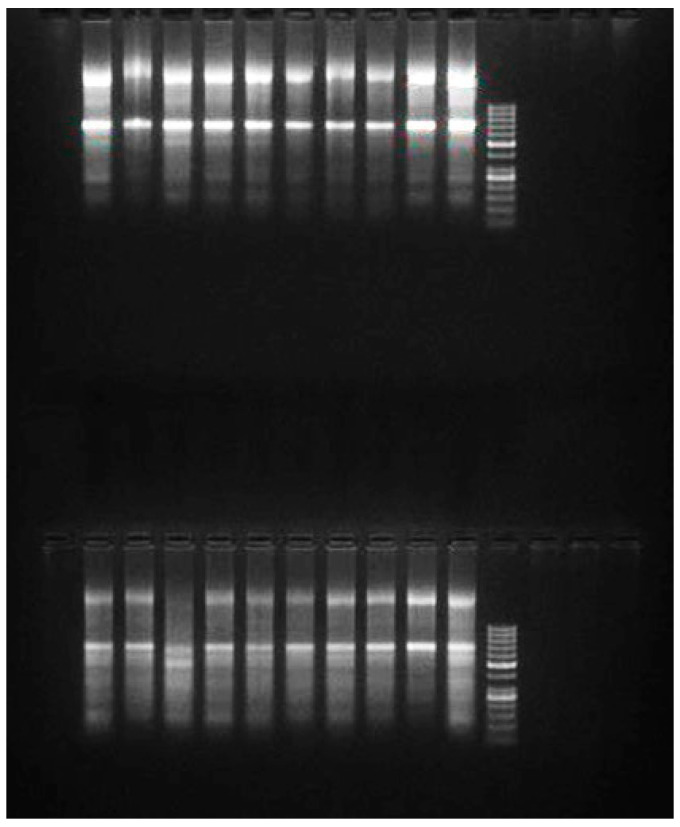
RNA image in 1% *w*/*v* agarose gel.

**Figure 2 animals-15-00016-f002:**
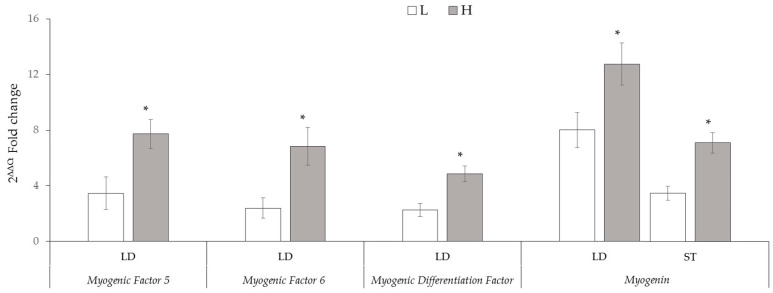
Expression levels of the *Myogenic Factor 5*, *Myogenic Factor 6*, *Myogenic Differentiation Factor,* and *Myogenin* genes in Longissimus-dorsi (LD) and Semitendinosus (ST) skeletal muscle of low-slaughter-weight (L) and high-slaughter-weight (H) Saanen kids. * *p* < 0.05.

**Table 1 animals-15-00016-t001:** Nutrient contents of experimental feeds (% on dry matter basis).

Nutrient	Concentrate	Alfalfa Hay
Dry matter	91.05	93.98
Crude protein	18.42	14.87
Crude fiber	6.38	25.05
Crude fat	3.71	0.71
Crude ash	7.28	10.11
ME (kcal/kg dry matter) *	2801.43	1863.00

* Metabolizable energy (ME) values of concentrate feed and alfalfa hay were calculated by following prediction equations described by Alderman [15] (ME (kcal/kg dry matter) = ((11.78 + (0.00654 × crude protein %) + (0.000665 × crude fat %)^2^ − (0.00414 × crude fat %) × crude fiber % − (0.0118 × crude ash %))/4.184) × 1000), and Menke & Steingass [16] (ME (kcal/kg dry matter) = 1.68 + (0.1418 × gasses production) + (0.073 × crude protein %) + (0.217 × crude fat %) − (0.025 × crude ash %) × 1000), respectively.

**Table 2 animals-15-00016-t002:** Primer sequences for the mRNA expression analysis of genes.

Genes	Primer Sequence (5′–3′)	Product Size (bp)	GenBank
*Myf5*	F-CACGACCAACCCTAACCAGAG R-TCTCCACCTGTTCCCTTAGCA	129	JF829004
*Myf6*	F-CGGAGCGCCATTAACTACAT R-AAATCCGCACCCTCAAGATT	113	NM_001285602
*MyoD*	F-GCCTGAGCAAAGTCAACGAG R-GAGTCGCCGCTGTAGTGTTC	72	JF829005
*MyoG*	F-GCAGCGCCATCCAGTACATAG R-GAAGGCCGCAGTGACATCC	124	JF829006
*GAPDH*	F-GCAAGTTCCACGGCACAG R-TCAGCACCAGCATCACCC	79	AF035421

**Table 3 animals-15-00016-t003:** Body weight characteristics of low-slaughter-weight (L) and high-slaughter-weight (H) Saanen kids at various ages.

Traits (g)	Groups			
	L (*n* = 11)	H (*n* = 13)	SEM	Sig.
Birth weight	2396	2580	37.31	0.009
Weaning weight at 120 day	18,280	19,820	380.73	0.039
Body weight at 140 day	24,520	25,850	386.21	0.185
Body weight at 150 day	26,000	27,210	333.15	0.167
Body weight at 160 day	26,530	28,070	310.68	0.009
Body weight at 170 day	27,360	30,130	311.85	0.001
Body weight at 180 day	27,730	31,175	456.13	0.001
DWG from birth to weaning	105.89	114.93	2.37	0.146
DWG from weaning to 180 days	157.50	189.25	5.27	0.001

Sig. = *p*-value of significant differences, SEM = standard error of mean, DWG = daily weight gain.

**Table 4 animals-15-00016-t004:** Carcass and non-carcass components characteristics of low-slaughter-weight (L) and high-slaughter-weight (H) Saanen kids.

Traits	Groups			
	L (*n* = 11)	H (*n* = 13)	SEM	Sig.
Carcass weight (g)				
Warm	14,064.56	15,558.44	274.03	0.001
Cold	13,677.41	15,164.84	282.24	0.001
Chilling loss	387.62	393.24	35.30	0.893
Carcass yield (%)				
Warm	50.35	49.49	0.47	0.144
Cold	48.96	48.23	0.52	0.231
Chilling loss	1.39	1.26	0.13	0.375
Non-carcass parts (g)				
Head	2186.66	2459.81	53.15	0.001
Feet	864.02	1012.84	20.64	0.001
Pelt	1960.02	2325.19	39.17	0.001
Kidney fat	85.74	81.34	6.50	0.552
Internal fat	185.08	187.18	12.97	0.886
Testis	222.76	214.38	5.75	0.262
Small intestine	1603.74	1872.44	83.76	0.013
Reticulo rumen	722.10	806.84	27.23	0.098
Organ weight (g)				
Kidney	97.01	118.28	2.98	0.001
Liver	450.10	533.72	14.86	0.001
Spleen	34.28	42.8	0.83	0.001
Heart	113.35	133.58	3.86	0.001
Lung	238.25	273.82	7.53	0.001

Sig. = *p*-value of significant differences, SEM = standard error of mean.

**Table 5 animals-15-00016-t005:** Carcass parts characteristics of low-slaughter-weight (L) and high-slaughter-weight (H) Saanen kids.

Traits		Groups			
		L (*n* = 11)	H (*n* = 13)	SEM	Sig.
Shoulder	g	3015.60	3308.00	45.90	0.001
	%	22.07	21.83	0.24	0.395
Flank	g	1470.40	1617.40	18.34	0.001
	%	10.77	10.67	0.15	0.368
Hind leg	g	5017.60	5804.20	152.30	0.011
	%	36.65	38.25	0.42	0.043
Loin	g	2013.60	2191.60	35.93	0.038
	%	14.73	14.46	0.13	0.645
Rib	g	1388.20	1608.40	42.75	0.001
	%	10.14	10.60	0.13	0.048
Neck	g	720.40	797.40	15.08	0.037
	%	5.27	5.26	0.04	0.724
Tail	g	42.00	49.00	1.22	0.001
	%	0.31	0.33	0.003	0.041

% = percentage for carcass weight. Sig. = *p*-value of significant differences, SEM = standard error of mean.

**Table 6 animals-15-00016-t006:** Carcass dissection characteristics of low-slaughter-weight (L) and high-slaughter-weight (H) Saanen kids.

Traits		Groups			
		L (*n* = 11)	H (*n* = 13)	SEM	Sig.
Meat (including fats and other tissue)					
Shoulder kol	g	2254.00	2499.20	37.36	0.021
	%	74.74	75.54	0.12	0.013
Flank etek	g	1143.20	1283.00	19.52	0.016
	%	77.74	79.30	0.40	0.025
Hind leg but	g	3773.40	4408.00	122.82	0.001
	%	75.18	75.92	0.16	0.048
Loin sırt bel	g	1510.36	1663.40	29.68	0.009
	%	74.99	75.91	0.33	0.063
Rib omuz	g	1004.60	1174.00	33.96	0.043
	%	72.33	72.96	0.22	0.578
Neck boyun	g	542.20	601.20	11.51	0.045
	%	75.26	75.40	0.09	0.643
Tail kuyruk	g	25.60	30.60	0.79	0.031
	%	60.92	62.47	0.29	0.038
Bone					
Shoulder	g	761.60	808.00	8.71	0.027
	%	25.26	24.44	0.12	0.043
Flank	g	327.20	339.40	4.95	0.069
	%	22.26	21.00	0.40	0.086
Hind leg	g	1244.20	1395.20	29.55	0.010
	%	24.82	24.07	0.16	0.048
Loin	g	503.20	518.00	4.14	0.148
	%	25.01	23.65	0.23	0.154
Rib	g	383.40	433.80	8.73	0.011
	%	27.65	27.00	0.22	0.548
Neck	g	178.20	194.80	3.26	0.028
	%	24.74	24.43	0.06	0.648
Tail	g	16.20	18.40	0.45	0.023
	%	38.60	37.53	0.23	0.041
Total meat	g	10,253.35	11,659.40	125.66	0.017
	%	74.95	76.86	0.51	0.039
Total bone	g	3414.00	3707.60	52.22	0.021
	%	24.98	24.47	0.24	0.648
Total meat-to-bone ratio		3.00	3.14	0.03	0.049

% = percentage for carcass weight. Sig. = *p*-value of significant differences, SEM = standard error of mean.

**Table 7 animals-15-00016-t007:** Correlation coefficients among the slaughter weight, carcass characteristics, and MRFs gene expression.

Measurements	Gene Expression (LD)				Gene Expression (ST)			
	*Myf5*	*Myf6*	*MyoD*	*MyoG*	*Myf5*	*Myf6*	*MyoD*	*MyoG*
SW	0.788 *	0.694 *	0.813 *	0.826 *	0.598 *	0.111	0.045	0.726 *
CCW	0.755 *	0.735 *	0.788 *	0.863 *	0.684 *	0.211	0.360	0.773 *
LW	0.782 *	0.641 *	0.891 *	0.706 *	-	-	-	-
HL	-	-	-	-	0.610 *	0.017	0.322	0.705 *
TMC	0.796 *	0.679 *	0.878 *	0.765 *	0.643 *	0.104	0.314	0.794 *

* *p* < 0.05, SW = slaughter weight, CCW = cold carcass weight, LW = loin weight, HL = hind leg weight, TMC = the total edible meat content of carcasses.

## Data Availability

The data presented in this study are available on request from the corresponding author.

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
