# Peer review of "Relationships Among Growth, Carcass Characteristics, and *Myf5 Myf6*, *MyoD,* and *MyoG* Genes Expression Level in Saanen Male Kids with Varying Slaughter Weights"

_animals, 2024, doi:10.3390/ani15010016_

Round 1
Reviewer 1 Report
Comments and Suggestions for Authors
1. Regarding the formatting issue of gene names: In the field of biology, in order to follow internationally recognized naming conventions, gene names should usually be represented in italics. This standard helps readers to more clearly identify the differences between genes and other terms, thereby improving the readability and professionalism of the paper. Please carefully review the entire text and change all gene names to italic format.
2. P-value format, in academic papers, it is usually recommended to italicize p-values in order to clearly distinguish statistical measures (such as p-values) from other textual content. Please review the entire text and change all p-values indicating significance (such as p<0.05, etc.) to italic format. While modifying the p-value format, please also ensure that the format of other statistics and variables conforms to the writing standards in the relevant field to maintain consistency in the format of the entire text.
3. When referring to genes for the first time, the full name should be used, and the abbreviation should be used throughout subsequent articles, such as in Figure 1 where "myogenic factor 5" should be used directly.
4. Lines 117-120 suggest providing a detailed description of what this study will do and what issues it will reveal.
5. Does line 164-165 comply with ethical and moral requirements.
6. It is suggested that the author provide the electrophoretic map of agarose gel electrophoresis to check the integrity of RNA.
7. The description of Table 3 in lines 229-231 is completely consistent. This repetitive description not only occupies unnecessary space, but may also confuse readers as to why there are two identical table explanations. It is recommended to delete one of the repeated table descriptions to maintain the compactness and coherence of the paper. Please carefully review the entire text to ensure that there are no other similar repetitive content or unnecessary redundant information.
8. Is there any literature support for lines 363-368.
9. It is suggested to delete lines 369-373.
10. What is the reason for the findings of this study in lines 374-376? Suggest citing relevant literature to support your findings.
11. In the results section of the paper, multiple research findings are closely connected but lack clear distinctions, making it difficult to quickly grasp the core content and relationships between each result when reading. To improve the readability and organization of the paper, it is recommended to add a subheading for each main result in the Results section.
12. The conclusion of the entire text needs to be reorganized. The conclusion section should provide a more comprehensive summary of the main findings of the study, ensuring that all important research results are mentioned.
13. Please carefully check the references to ensure their accuracy and necessity.
Author Response
Dear Reviewer,
Thank you very much for taking the time to review this manuscript (animals-3197824) entitled “Relationships among Growth, Carcass Characteristics and Myogenic Regulatory Factors Gene Expression Level in Saanen Male Kids with Varying Slaughter Weights." Please find the detailed responses below and the corresponding revisions/corrections highlighted/in track changes in the re-submitted files.
Reviewer Comments to Author and Author Response:
Comments 1: Regarding the formatting issue of gene names: In the field of biology, in order to follow internationally recognized naming conventions, gene names should usually be represented in italics. This standard helps readers to more clearly identify the differences between genes and other terms, thereby improving the readability and professionalism of the paper. Please carefully review the entire text and change all gene names to italic format.
Response 1: Done as requested. Gene names have been changed to italic format; please see lines 32-34, 46-49, 101-104, 108-110, 126, 231, 245-246, 313-320 in the revised MS. in the revised MS.
Comments 2: P-value format, in academic papers, it is usually recommended to italicize p-values in order to clearly distinguish statistical measures (such as p-values) from other textual content. Please review the entire text and change all p-values indicating significance (such as p<0.05, etc.) to italic format. While modifying the p-value format, please also ensure that the format of other statistics and variables conforms to the writing standards in the relevant field to maintain consistency in the format of the entire text.
Response 2: Done as requested. P-values have been changed to italic format; please see lines 231-262, 266-297, and 351-256 in the revised MS. in the revised MS.
Comments 3: When referring to genes for the first time, the full name should be used, and the abbreviation should be used throughout subsequent articles, such as in Figure 1 where "myogenic factor 5" should be used directly.
Response 3: Done as requested. The full name was used when referring to genes for the first time in the text, and the abbreviation was used subsequently; please see lines 101-103 and 106-110 in the revised MS. in the revised MS.
Comments 4: Lines 117-120 suggest providing a detailed description of what this study will do and what issues it will reveal.
Response 4: Done as requested. A detailed description has been added; please see lines 121-124 in the revised MS.
Comments 5: Does line 164-165 comply with ethical and moral requirements.
Response 5: Done as requested. All kids in the study were slaughtered according to standard commercial slaughter procedures in accordance with ethical and moral requirements. This information has been added to the revised MS; please see line 173.
Comments 6: It is suggested that the author provide the electrophoretic map of agarose gel electrophoresis to check the integrity of RNA.
Response 6: Done as requested. RNA agarose gel electrophoresis image has been added to the revised article; please see lines 210-226.
Comments 7: The description of Table 3 in lines 229-231 is completely consistent. This repetitive description not only occupies unnecessary space, but may also confuse readers as to why there are two identical table explanations. It is recommended to delete one of the repeated table descriptions to maintain the compactness and coherence of the paper. Please carefully review the entire text to ensure that there are no other similar repetitive content or unnecessary redundant information.
Response 7: Done as requested. The repetitive description has been deleted. Please see line 257 in the revised MS.
Comments 8: Is there any literature support for lines 363-368.
Response 8: Done as requested. Literatures have been added. Please see lines 375-377 in the revised MS.
Comments 9: It is suggested to delete lines 369-373.
Response 9: Done as requested. The sentence has been added. Please see lines 380-383 in the revised MS.
Comments 10: What is the reason for the findings of this study in lines 374-376? Suggest citing relevant literature to support your findings.
Response 10: Done as requested. The literature and a sentence have been added to support our findings. Please see lines 386-390 in the revised MS.
Comments 11: In the results section of the paper, multiple research findings are closely connected but lack clear distinctions, making it difficult to quickly grasp the core content and relationships between each result when reading. To improve the readability and organization of the paper, it is recommended to add a subheading for each main result in the Results section.
Response 11: Done as requested. Subheadings for each main result in the results section have been added. Please see lines 256, 269, 312, and 347 in the revised MS.
Comments 12: The conclusion of the entire text needs to be reorganized. The conclusion section should provide a more comprehensive summary of the main findings of the study, ensuring that all important research results are mentioned.
Response 12: Done as requested. The conclusion has been revised. Please see lines 514-518 in the revised MS.
Comments 13. Please carefully check the references to ensure their accuracy and necessity.
Response 13: Done as requested. All references have been checked in the revised MS.
Regards

Reviewer 2 Report
Comments and Suggestions for Authors
The paper dug into the relationship between the expression levels of MRFs gene family members (Myf5, Myf6, MyoD, and MyoG) and growth and carcass characteristics in Saanen male kids of the same age with varying slaughter weights. The result showed that the difference in slaughter weight at the end of fattening in Saanen kids may be due to the expression level of MRFs genes. And illustrated the fattening performance and MRFs genes may have a positive correlation. This study is of great significance for determining the factors affecting the fattening quality of male lambs of dairy goats, improving carcass quality and increasing productivity.
But there are some problems, which must be solved before it is considered for publication:
According to the materials and methods in this paper, lambs were weaned at 5 months and fattened intensively for 60d, while the weaning age indicated by the author in Table 3 of the results and analysis section was 120d. Therefore, whether the expression of "5 months" here was correct needs to be checked and modified.
And there are several details in the article that need to be reviewed, such as repeated statements(the statements in lines 230 and 231 are repetitive and should be modified),and the significant letter "p" needs to be written in italics.
Author Response
Dear Reviewer,
Thank you very much for taking the time to review this manuscript (animals-3197824) entitled “Relationships among Growth, Carcass Characteristics and Myogenic Regulatory Factors Gene Expression Level in Saanen Male Kids with Varying Slaughter Weights." Please find the detailed responses below and the corresponding revisions/corrections highlighted/in track changes in the re-submitted files.
Reviewer Comments to Author and Author Response:
Comments 1: According to the materials and methods in this paper, lambs were weaned at 5 months and fattened intensively for 60d, while the weaning age indicated by the author in Table 3 of the results and analysis section was 120d. Therefore, whether the expression of "5 months" here was correct needs to be checked and modified.
Response 1: Done as requested. The spelling mistake has been corrected. Please see lines 40-42 in the revised MS.
Comments 2: And there are several details in the article that need to be reviewed, such as repeated statements (the statements in lines 230 and 231 are repetitive and should be modified), and the significant letter "p" needs to be written in italics.
Response 2: Done as requested. The repetitive description has been deleted. Please see line 257 in the revised MS. P-values have been changed to italic format; please see lines 231-262, 266-297, 351-256 in the revised MS. in the revised MS.
Regards

Reviewer 3 Report
Comments and Suggestions for Authors
Abstract
Line 37-38 can be deleted and repeat the same information as the lines below
What is the objective of the study?
Introduction
This is too long and does not clearly present the objective and hypothesis of the study
Discussion
The big problem with the discussion is that it is aimed at discussing the difference in the expression of MFRs as the factor responsible for changes in carcass parameters; however, the difference in the expression of MFRs is a response to treatments
Conclusion
The same advice is given to the discussion
Author Response
Dear Reviewer,
Thank you very much for taking the time to review this manuscript (animals-3197824) entitled “Relationships among Growth, Carcass Characteristics and Myogenic Regulatory Factors Gene Expression Level in Saanen Male Kids with Varying Slaughter Weights." Please find the detailed responses below and the corresponding revisions/corrections highlighted/in track changes in the re-submitted files.
Reviewer Comments to Author and Author Response:
Comments 1: Abstract; Line 37-38 can be deleted and repeat the same information as the lines below.
Response 1: Done as requested. The repetitive information has been deleted. Please see lines 40-43 in the revised MS.
Comments 2: Abstract; What is the objective of the study?
Response 2: Done as requested. The objective of the study has been added. Please see lines 38-39 in the revised MS.
Comments 3: Introduction; This is too long and does not clearly present the objective and hypothesis of the study
Response 3: Done as requested. The introduction was shortened, and sections that did not clearly support the objective and hypothesis of the study were removed. Please see lines 57-72 in the revised MS.
Comments 3: Discussion; The big problem with the discussion is that it is aimed at discussing the difference in the expression of MFRs as the factor responsible for changes in carcass parameters; however, the difference in the expression of MFRs is a response to treatments.
Response 3: Thank you for pointing this out. We agree with this comment, but the experimental animals were grown and fattened under similar conditions, so no treatment was applied. Therefore, the discussion of the article was to relate the reason for the different slaughter weights of the animals grown and fattened under the same conditions to the expression level of the MRFs gene family.
Comments 4: Conclusion; The same advice is given to the discussion
Response 4: Done as requested. The conclusion has been revised. Please see lines 514-518 in the revised MS.

Round 2
Reviewer 1 Report
Comments and Suggestions for Authors
The author has made comprehensive revisions to the issues raised last time, and the current version is acceptable
Author Response
Thank you!
Reviewer 3 Report
Comments and Suggestions for Authors
The big problem to be solved in the study is the discussion, the authors only included references, the results must be discussed and this is not happening.
Author Response
The research findings were discussed by evaluating the results of the previous studies in the article. The referee's statement, "The big problem to be solved in the study is the discussion, the authors only included references, the results must be discussed and this is not happening." was not very well understood by us, but a few additions were made to the discussion section of the article to improve it (please see lines 398-401, 410-412). If there are any deficiencies in the article's discussion section, we are willing to correct them if deficiencies are clearly stated.
Round 3
Reviewer 3 Report
Comments and Suggestions for Authors
no comment
Author Response
Dear Reviewer,
Thank you very much for taking the third time to review this manuscript (animals-3197824) entitled “Relationships among Growth, Carcass Characteristics and Myogenic Regulatory Factors Gene Expression Level in Saanen Male Kids with Varying Slaughter Weights." Please find the detailed responses below and the corresponding revisions/corrections highlighted/in track changes in the re-submitted files.
Reviewer(s)' Comments to Author:
Reviewer #1:
Comments 1: Title, what does 'Myogenic Regulatory Factors Gene Expression Level' mean? The title must be intelligible to the average reader.
Response 1: Done as requested. The title has been revised to be intelligible to the average reader in the revised MS.
Comments 2: L 30,31 repetition of therefore.
Response 2: Done as requested. The repetition of "therefore" was deleted; please see line 32 in the revised MS.
Comments 3: L 41 were divided.
Response 3: Done as requested. “were” added to text; please see line 42 in the revised MS.
Comments 4: L 50 and L 498 you have shown an association, but not causation. This conclusion is overstated.
Response 4: Done as requested. Sentences have been added to explain the reason for the association; please see lines 50-51 and 517-519 in the revised MS.
Comments 5: L 55 delete thus.
Response 5: Done as requested. “thus” deleted to text; please see line 57.
Comments 6: L 58 delete number.
Response 6: Done as requested. “number” was deleted from the text; please see line 60.
Comments 7: L 59 Therefore, the low financial value of surplus male kids also increases potential welfare concerns in the dairy goat industry why? it is not clear. You give some sort of explanation in the subsequent sentences, but fail to draw the obvious conclusion, surplus male kids are not well looked after, not well fed, not given quality accommodation etc.
Response 7: Done as requested. The sentence was revised, and more descriptive terms have been added; please see lines 60-61 in the revised MS.
Comments 8: L 113 why were they dried, were they wet?
Response 8: Since the kids were weighed immediately after birth, the placental fluid (amniotic fluid) on them was dried and weighed to determine their actual birth weight.
Comments 9: L 124 describe the reasons
Response 9: Done as requested. A sentence explaining the reasons why the kids died was added. Please see lines 126-127 in the revised MS.
Comments 10: L 155 describe the processes. Were they stunned before slaughter? If not it raises ethical questions.
Response 10: Done as requested. All the kids were stunned before slaughter. This information was added. Please see lines 162-163 in the revised MS.
Comments 11: Table 3 189,25 should be 189.25.
Response 11: Done as requested. “189,25” was corrected as “189.25”; please see Table 3 in the revised MS.
Comments 12: Statistical analysis, did you test for the assumptions of ANOVA, homogeneity of variance, normality of residuals. If not, do so. Only one error estimate per parameter is needed and it is not necessary to use superscript letters to how that means in rows with different letters are significantly different at p<0.05. But you must give a P value for each parameter and a single error estimate, SED.
Response 12: Done as requested. Information was added to the statistical analysis section; please see lines 246-247 in the revised MS. Also, P values were added to tables, and the standard error of means was pooled for each trait.
Comments 13. Please review the number of significant figures for means, only state to precision with which you can measure, in some places you use 5 or 6 SF.
Response 13: Done as requested. Figure 2 was revised. Please see lines 327-347 in the revised MS.
